# Association of a Novel Medication Risk Score with Adverse Drug Events and Other Pertinent Outcomes Among Participants of the Programs of All-Inclusive Care for the Elderly

**DOI:** 10.3390/pharmacy8020087

**Published:** 2020-05-20

**Authors:** David L. Bankes, Hubert Jin, Stephanie Finnel, Veronique Michaud, Calvin H. Knowlton, Jacques Turgeon, Alan Stein

**Affiliations:** 1Applied Precision Pharmacotherapy Institute, Tabula Rasa HealthCare, Moorestown, NJ 08057, USA; DBankes@trhc.com; 2Healthcare Analytics, Tabula Rasa HealthCare, Moorestown, NJ 08057, USA; HJin@trhc.com (H.J.); SFinnel@trhc.com (S.F.); 3Precision Pharmacotherapy Research and Development Institute, Tabula Rasa HealthCare, Lake Nona, Orlando, FL 32827, USA; VMichaud@trhc.com (V.M.); JTurgeon@trhc.com (J.T.); 4Chief Executive Officer, Tabula Rasa HealthCare, Moorestown, NJ 08057, USA; CKnowlton@trhc.com

**Keywords:** Programs of All-Inclusive Care for the Elderly, PACE, medical expenditures, medication risk stratification, medicare, adverse drug events, risk score

## Abstract

Preventable adverse drug events (ADEs) represent a significant public health challenge for the older adult population, since they are associated with higher medical expenditures and more hospitalizations and emergency department (ED) visits. This study examines whether a novel medication risk prediction tool, the MedWise Risk Score™ (MRS), is associated with ADEs and other pertinent outcomes in participants of the Programs of All-Inclusive Care for the Elderly (PACE). Unlike other risk predictors, this tool produces actionable information that pharmacists can easily use to reduce ADE risk. This was a retrospective cross-sectional study that analyzed administrative medical claims data of 1965 PACE participants in 2018. To detect ADEs, we identified all claims that had ADE-related International Classification of Diseases and Health Related Problems, 10th revision (ICD-10) codes. Using logistic and linear regression models, we examined the association between the MRS and a variety of outcomes, including the number of PACE participants with an ADE, total medical expenditures, ED visits, hospitalizations, and hospital length of stay. We found significant associations for every outcome. Specifically, every point increase in the MRS corresponded to an 8.6% increase in the odds of having one or more ADEs per year (OR = 1.086, 95% CI: 1.060, 1.113), $1037 USD in additional annual medical spending (adjusted R^2^ of 0.739; *p* < 0.001), 3.2 additional ED visits per 100 participants per year (adjusted R^2^ of 0.568; *p* < 0.001), and 2.1 additional hospitalizations per 100 participants per year (adjusted R^2^ of 0.804; *p* < 0.001). Therefore, the MRS can risk stratify PACE participants and predict a host of important and relevant outcomes pertaining to medication-related morbidity.

## 1. Introduction

Among older Americans, adverse drug events (ADEs) represent a significant public health challenge. Estimates indicate that ADEs may occur in up to 35% of older outpatients [1] and cause or contribute to 6%–12% of this cohort’s hospital admissions [2]. Predictably, ADEs carry economic consequences. Medication-related morbidity and mortality may be responsible for 16% of the United States’ annual healthcare costs, representing a $500 billion annual toll [3]. Fortunately, the US healthcare system should be able to mitigate the economic and clinical consequences of ADEs. Consistent data suggest that the majority of ADEs are both predictable and preventable [1,4].

The predictable and preventable nature of ADEs is a call to action. A risk stratification tool that identifies individuals at risk could help policy makers, healthcare organizations, and payers curb the negative clinical and economic consequences of ADEs. While ADE risk prediction tools do exist, a disproportionate amount have been validated in hospital settings and, as recently reported, several of them cannot accurately predict the risk of drug-related problems [2,5,6]. Moreover, many of the existing tools utilize chronic comorbidities or laboratory values, such as renal failure, heart failure, or white blood cell count, to predict ADEs [5,7,8]. While these variables may accurately predict ADE risk, they are not necessarily modifiable or easily actionable. Thus, they are of dubious clinical utility. By contrast, the MedWise Risk Score™ (MRS) [9], described by Cicali et al., uses potentially alterable pharmacological characteristics of patients’ medications for risk stratification, making it potentially easier for healthcare providers to mitigate medication risk. However, until now, associations with ADEs and other important outcomes (e.g., medical expenditures, emergency department visits, hospitalizations) have not been reported in controlled studies.

The US-government funded Program of All-Inclusive Care for the Elderly (PACE) is an ideal setting to examine the MRS. PACE provides supportive services to community-dwelling older adults who are certified by the state in which they reside to require a “nursing home level of care” [10]. Given a capitated payment model, PACE organizations aim to avoid unnecessary medical expenditures, nursing home institutionalization, and frequent hospital visits [10]. PACE organizations are at great risk of unnecessary spending associated with medication-related morbidity given their participants’ significant comorbidities [11] and prevalence of medication-related problems [12]. A tool like the MRS could identify these organizations’ high-risk members and provide guidance to mitigate risk. Therefore, our objective here is to examine whether the MRS correlates with ADEs as well as the outcomes that are important to PACE and other government-funded healthcare programs: total cost, emergency department (ED) visits, hospitalizations, and hospital length of stay.

## 2. Materials and Methods

### 2.1. Study Design, Setting, and Context

This was a retrospective cross-sectional study that analyzed administrative medical claims data of PACE participants who received care from a national provider of PACE pharmacy services, CareKinesis. This study was granted a waiver of informed consent by the Biomedical Research Alliance of New York Institutional Review Board (protocol number 19-12-172-427, approved 24 May 2019). Researchers conducted the study in accordance with ethical principles set forth by the Declaration of Helsinki.

The PACE program has been described extensively in previous publications [10,13]. To highlight, each PACE organization receives capitated funding from the Centers for Medicare and Medicaid Services (CMS) and their state’s Medicaid program to provide comprehensive, supportive services to a unique cohort of medically-complex, community-dwelling older adults. The main objective of PACE is to prolong independent living within the community. Given a capitated payment model, PACE organizations have discretion to include any needed service to help promote favorable outcomes. PACE patients, known as “participants,” have their care coordinated by an interdisciplinary team (IDT) of professionals (e.g., primary care physicians, social workers, nursing, physical therapy, etc.). Participants travel to day-care centers to receive medical and supportive services as determined by the IDT. At the time of writing, there are over 250 PACE centers in 31 states in the continental US [11]. Each center often partners with one pharmacy for prescriptions and consultative services [13,14].

As of this writing, CareKinesis PACE Pharmacy provides comprehensive medication services to more than 14,000 PACE participants across the continental US. This census represents nearly 100 individual PACE centers. In addition to providing prescriptions and over-the-counter medications, the pharmacy offers a suite of clinical services delivered by board-certified geriatric pharmacists (i.e., the BCGP credential) who use a novel clinical decision support system (CDSS; EireneRx^®^). The CDSS helps pharmacists identify and resolve specific medication-related problems [12] as well as optimize medication regimens based on available pharmacogenomic information [15]. Prescribers and pharmacists commonly collaborate to improve medication safety and reduce ADE risk [12,16].

### 2.2. Study Sample

This study attempted to assess associations between the MRS and outcomes that occur on an annual basis. As a result, the investigators limited the analytical study sample to PACE participants with complete (12 months) 2018 medical claims (physician and facility charges) data and 12 months of MRS (i.e., risk stratification) data during the same period. Therefore, we excluded PACE participants for whom we did not have access to medical claims data (*n* = 9689), did not have risk stratification data (*n* = 85), and did not have 12 months of 2018 claims and/or risk stratification data (*n* = 249).

### 2.3. Definitions, Terms, and Measures

#### 2.3.1. The Novel MRS

Within the CDSS, MRS is calculated in real time for every participant; therefore, the MRS is updated each time the medication regimen changes. The MRS is generated by processing patient medication data (e.g., National Drug Codes, RxNorm’s Rx Concept Unique Identifiers) within the e-prescribing and pharmacy management platform (EireneRx^®^). For both prescribers and pharmacists, the MRS is displayed prominently at the top of every participant’s medication profile. Moreover, “sandbox” features help clinicians understand how hypothetical drug regimen alterations (e.g., prescription or over-the-counter medication additions or deletions) can impact the participant’s MRS. This helps the prescriber or pharmacist make informed decisions before making or suggesting changes to the medication regimen. PACE prescribers receive accredited education regarding the MRS.

The MRS is the object of a patent currently under review (WO 2019/089725; pending) and has been previously described in the literature [9]. Briefly, it is derived from aggregated, weighted values for five pharmacokinetic and pharmacodynamic factors shown to be associated with medication-related morbidity and mortality. The total MRS ranges from 0 to 53. Overall, the factors comprising this model have been tested and there is minimal observed multicollinearity. Some collinearity does exist between factors 2 and 3, but the magnitude of collinearity is dependent upon the cohort’s sample size. All five risk factors are further recapped below along with a brief justification for inclusion in the MRS.

By assigning relative odds ratios (RORs) [17] for all single-drug adverse event pairs, factor 1 accounts for the risk of ADEs as reported to the US Food and Drug Administration’s (FDA) Adverse Event Reporting System (FAERS) database. In our analysis, odds ratios (ORs) are calculated for 104 adverse events for each active ingredient. Next, a ROR is calculated by accounting for the entire drug regimen. For example, if a participant has four drugs associated with the same ADE in his/her drug regimen, the calculated ROR accounts for the cumulative frequency of each OR, weighted by the total number of observations in the FAERS for these active ingredients. Factor 2 utilizes the Anticholinergic Cognitive Burden (ACB) scale to calculate the drug regimen’s total anticholinergic burden [18,19]. Cumulative use of anticholinergic medications may contribute to higher total healthcare utilization [20], chronic functional decline, cognitive deficits, weakness, falls, hospitalizations, and all-cause mortality [5,21,22,23,24,25,26]. Factor 3 utilizes the Sedative Load (SL) model [27] of each medication to quantify the sedative burden of the drug regimen. Exposure to highly sedating medications has been associated with poor patient outcomes and adverse drug events, such as cognitive decline, physical impairment, falls, and increased mortality [28,29,30,31]. Factor 4 quantifies the medication regimen’s risk of QT-prolongation because studies have shown a positive correlation between increased QTc length and mortality [32,33]. The QT interval is the time from the start of the Q wave to the end of the T wave on the surface electrocardiogram. The QTc is this interval corrected for heart rate. Lastly, factor 5 utilizes drug metabolizing information and, when available, pharmacogenomic data to quantify the number, severity, and complexity of pharmacokinetic drug interactions. Drug interactions are associated with ADE-related hospitalizations [34,35] and, when available, pharmacogenomic data can help clinicians identify more clinically significant interactions [36].

Since the MRS is calculated in real time for every PACE participant’s drug regimen, the MRS can account for acute-on-chronic medication risk. For example, a pharmacokinetic drug interaction is expected in a patient taking chronic warfarin who is prescribed acute antibiotic therapy with sulfamethoxazole-trimethoprim, resulting in a higher risk of hemorrhaging [37]. Therefore, this will yield a higher MRS compared to baseline from warfarin use alone. To capture acute-on-chronic medication risk and/or worsening risk due to changes in chronic drug therapy, the participant’s highest reported MRS during 2018 was identified to perform all analyses in this study. This value captures the temporal nature of medication risk.

#### 2.3.2. ADEs

To detect ADEs, we utilized a list of ADE-related International Classification of Diseases and Health Related Problems, 10th revision (ICD-10) codes classified by Hohl et al. [38]. In this study, the authors identified 827 ICD-10 codes that have been used in the literature to identify ADEs in claims data. They classified these codes into nine categories, based on the likelihood that each code indicated the presence of an ADE: A1: “induced by medication/drug”; A2: “induced by medication or other causes”; B1: “poisoning by medication”; B2: “poisoning by or harmful use of medication or other causes”; C: “ADE very likely; D: “ADE likely”; E: “ADE possible”; U: “ADE unlikely”; V: “ADE related to vaccine”. We limited our analysis to 325 codes that were classified as A1, A2, B1, and B2 because these codes unambiguously indicate the presence of an ADE. Therefore, our analysis reflects a conservative view of ADEs. To avoid overcounting redundant ADE claims (e.g., multiple similar ADE-related ICD-10 codes recorded over the course of consecutive days or in close proximity), we did not count the total number of ADEs per member. Rather, we considered ADEs to be a binary variable that was either present or absent for a given participant. Specifically, if a participant had evidence of ≥1 ADE-related ICD-10 code in 2018 claims, the participant would be considered “ADE positive”. Conversely, if there was no ADE-related ICD-10 code in the claims, the participant was considered “ADE negative”.

#### 2.3.3. Pertinent Risk Outcomes

In addition to ADEs, we assessed the association between the MRS and several other outcomes that have been associated with ADEs: total annual combined facility and physician expenditures, all-cause emergency department (ED) visits, all-cause hospital admissions, and hospital length of stay. Facility expenditures were defined as the total adjudicated amount (USD) for a participant’s care received from hospitals and/or skilled nursing facilities. Physician expenditures were defined as the total adjudicated amount (USD) from physician charges for participant care. ED visits and hospitalizations were identified by their line-item claims. Hospital length of stay was the number of consecutive days between the admission date and the discharge date, where these two dates were identified based on claims.

### 2.4. Statistical Procedures

We utilized a binary logistic regression model to determine the probability of having at least 1 ADE (i.e., participant had either ≥1 ADE or no ADE) at any given MRS. For a more intuitive analysis and interpretation, we also utilized a linear regression to evaluate the association between the MRS and the number of members with at least one ADE in 2018. The regression model was weighted by the number of members at each risk score point to diminish any impact of outliers. To determine the MRS’ ability to predict ADE presence, a receiver operator characteristic (ROC) curve was constructed, and the area under the curve (AUC) was calculated. For the other outcomes, relationships were examined using linear regression models, again weighted by the number of members at each risk score point. An alpha level of 0.05 was defined a priori to determine statistical significance. All analyses were processed using R version 3.5.3.

## 3. Results

During 2018, 11,988 patients were identified in CareKinesis’ total census. After making various exclusions, which are outlined in Figure 1, 1965 PACE participants remained in the analytical sample. PACE participants included in this analysis were 65.2% (*n* = 1282) female and, on average 76.8 ± 9.9 years old. Participants represented a total of 12 PACE organizations, which are geographically dispersed throughout the United States and vary in overall census size. Table 1 highlights participant demographics. Figure 2 depicts the overall distribution of the MRS among PACE participants. Overall, MRS ranged from 2 to 40 in the total sample and, on average, participants had an MRS of 18.5 ± 7.8.

During 2018, 128 (6.5%) PACE participants experienced at least one ADE. A total of 54 unique ADE-related ICD-10 codes were identified in the 2018 claims. The ICD-10 codes that described ADEs were often related to opioids, extrapyramidal symptoms, anticoagulants, skin eruptions, and psychoactive medications. These symptoms and medication classes accounted for at least 50% (*n* = 85) of all (*n* = 170) ADE claims. During our analysis, we found patients with ADE codes for T78.4 (allergy, unspecified), T80.2 (infections following infusion, transfusion, or therapeutic injection), and T40.1 (heroin poisoning). These codes were excluded from our analysis because such ADEs are unrelated to the MRS. The top ADEs and their corresponding ICD-10 codes can be viewed in Table 2. All remaining ADE codes are provided in Appendix A, Table A1.

The logistic regression analysis found that the odds of having one or more ADEs over the course of the year increases by 8.6% per every point increase in the MRS (OR = 1.086, 95% CI: 1.060, 1.113; *p* < 0.001). A weighted linear regression was also calculated, using MRS to predict the total number of participants with at least one ADE. A significant regression equation was found (F (1, 36) = 31.8; *p* < 0.001), with an adjusted R^2^ of 0.454. The number of PACE participants predicted to have at least one annual ADE is equal to –0.0261 + 0.0049x, where x is the MRS. Therefore, every point increase in the MRS corresponds to an additional 4.9 participants per 1000 PACE participants with at least one ADE per year. The ADE vs. MRS weighted regression can be viewed in Figure 3. The area under the ROC curve, which quantifies the ability of the MRS to predict the presence of ADEs, was 0.67, and is depicted in Appendix A, Figure A1.

We also assessed the MRS’ relationship to a variety of other pertinent risk outcomes, which can be viewed in Figure 4, Figure 5, Figure 6 and Figure 7. Notably, a significant association was observed when MRS was regressed against total annual medical expenditures. Figure 4 depicts this weighted linear regression. A significant regression equation was found (F (1, 36) = 105.8; *p* < 0.001), with an adjusted R^2^ of 0.739. PACE participants predicted total annual facility and physician costs as equal to 7091.50 + 1036.90x USD, where x is the MRS. Therefore, PACE participants’ annual costs increase $1036.90 USD for every point of MRS. Figure 5, Figure 6 and Figure 7 show all other weighted regressions calculated to predict various risk outcomes based on MRS. In summary, we observed a significant positive correlation between the MRS and various outcomes, including annual ED visits, all-cause hospital admissions, and hospital length of stay.

## 4. Discussion

This retrospective study of PACE administrative medical claims demonstrates that a novel MRS derived strictly from a medication regimen’s pharmacological properties was significantly associated with ADE occurrence among medically-complex, community-dwelling older adults enrolled in a PACE program. Specifically, we found that every point increase in the MRS corresponded to nearly five additional participants having at least one ADE in the year out of every 1000 participants. However, this is likely a gross underestimation. The literature indicates that ADEs are widely underreported, with several studies citing underreporting rates that exceed 90% [39]. To account for potential underreporting and given established associations with higher healthcare utilization and cost, we anticipated that a higher MRS would be associated with higher costs, hospital admissions, ED visits, and hospital length of stay. Our results supported this idea, as each point increase in the MRS corresponded to over $1000 USD in additional annual medical spending, three additional annual ED visits per 100 participants per year, and two additional hospitalizations per 100 participants per year. Collectively, these results indicate that it is possible to utilize the pharmacological properties of a drug regimen to risk stratify PACE participants and predict a host of important and relevant outcomes pertaining to medication-related morbidity.

The associations identified in this study have important implications for PACE organizations, which are at full financial risk for participant outcomes due to capitated payments from CMS and state Medicaid programs [10]. Projections through 2025 estimate that, in general, costs of medical resources (e.g., hospitalizations and emergency room visits) will increase, while many states could face future annual Medicaid funding restrictions [3,13,40]. This combination will place immense pressure on PACE to engage in cost avoidance and reduction activities [13]. Such activities must not ignore medication-related morbidity, which could cost the healthcare system an additional $2500 USD for each instance of non-optimized drug therapy [3]. Given that resolution of medication-related problems may help avoid unnecessary medical expenditures [3,41], deploying consistent medication risk identification and mitigation strategies in PACE is vital.

While the utilization of such strategies remains inconsistent in this setting [10,13,14], some evidence indicates that PACE healthcare providers (HCPs) desire comprehensive and consistent support to help promote safer pharmacotherapy. Sloane et al. found that PACE physicians frequently reported difficulties in understanding how patients’ complex morbidity can alter risk of adverse health outcomes or wasted spending. This lack of understanding can lead to treatment that is inappropriate, ineffective, or unnecessary [42]. Physicians outside of PACE have echoed these sentiments regarding the pharmaceutical care of the medically-complex older adult. Specifically, polypharmacy resulting from multi-morbidity makes medication management in this cohort especially challenging [43,44,45]. Underscoring these challenges, a recent study found that PACE physicians and nurse practitioners frequently initiated consultation with pharmacists for information and advice when faced with medication safety uncertainties [16]. Notably, their inquiries commonly pertained to opioids and psychoactive drugs [16], which are medication classes responsible for the disproportionate amount of ADE claims identified in the present study.

The MRS can help solve these problems. First, PACE HCPs can now use the MRS to identify participants who are at the greatest risk of medication morbidity and healthcare utilization. Therefore, PACE organizations and pharmacists can properly allocate efforts and resources to the participants who have the greatest need for intervention. For instance, pharmacists and HCPs can focus initial medication reviews on 20% of the population at greatest risk. In this study, this would correspond to participants with an MRS ≥26. Second, point-of-care access to the MRS provides HCPs with actionable information to make geriatric medication management less challenging. The MRS is comprised of five risk factors that highlight aspects of the regimen that need the most intervention or attention. For example, if the MRS identifies that a PACE participant is at high risk due to competitive inhibition, HCPs can act to resolve any present interaction(s).

By identifying tangible ways to mitigate medication-related morbidity among the most “at risk” PACE participants, the MRS can profoundly influence HCP behavior. For example, a survey of PACE physicians found that the overwhelming majority (>85%) of respondents were more likely to deprescribe medications or reconsider medication choice as a result of having access to the MRS [46]. When PACE pharmacists were supported with the MRS in a CDSS, HCPs accepted about 80% of pharmacists’ recommendations to resolve medication-related problems that frequently involved broad medication safety concerns (69%), such as adverse drug reactions (18%) and pharmacokinetic drug interactions (21%) [12]. The interactions frequently involved opioids, anticoagulants, and psychoactive medications [12], which were top contributors to ADEs in the present study. Collectively, this suggests that the MRS helps PACE HCPs and pharmacists identify and mitigate relevant medication risk.

Our analysis is not without limitations, which carry pertinent future research directives. First, retrospectively detecting ADEs with a conservative set of ICD-10 codes may have biased our results and underestimated the true ADE occurrence. As aforementioned, administrative data have a low sensitivity to detect ADEs since it is well-established that ADEs are underreported by HCPs at the point of care [38,39,47]. Since ADEs are a difficult factor to capture through administrative claims alone, this could explain why our area under the ROC curve was fair at 0.67. To our knowledge, other risk scores only examined ADE-induced hospitalizations or inpatient-acquired ADEs, making comparisons difficult. Nevertheless, it is noteworthy that our metric was similar to the area under the ROC curve reported by Parameswaran Nair et al. (0.70), who designed a risk score to detect ADE-related hospitalizations among older outpatients whereby ADEs were clinically validated by expert consensus chart review and patient interviews [5]. Future studies should assess the MRS in a prospective manner, with clinical ADE validation. Nevertheless, associations with ADEs were still statistically significant despite using ICD-10 codes. Therefore, the associations observed between the MRS and the other negative risk outcomes can be explained, at least in part, by medication-related morbidity. Future research will need to validate the MRS against other ADE-specific risk scores [5,6] and various risk indices highly relevant to PACE (e.g., Hierarchical Condition Category scores). Finally, additional studies will be needed in alternative cohorts. Given the uniqueness of the PACE cohort, our results may not be generalizable to inpatient older adults and healthier community-dwelling older adults.

## 5. Conclusions

We found that a novel MRS that derives a level of risk strictly through aggregation of pharmacological properties of a medication list positively correlates with ADEs, total utilization, hospitalizations, ED visits, and hospital length of stay among PACE participants. Adopting the MRS in PACE offers PACE organizations the opportunity to identify and prospectively manage participants at risk of ADEs and other relevant outcomes in a standardized manner.

## Figures and Tables

**Figure 1 pharmacy-08-00087-f001:**
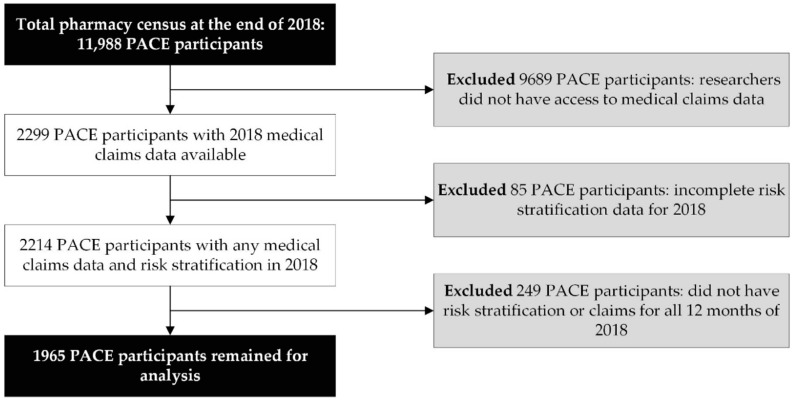
Data management and workflow diagram.

**Figure 2 pharmacy-08-00087-f002:**
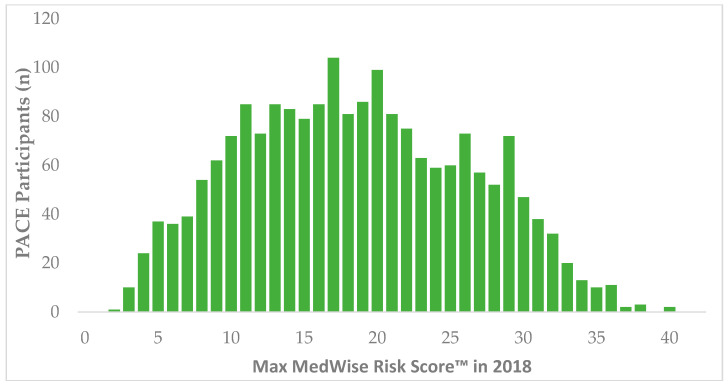
Distribution of MedWise Risk Score™ among PACE participants. Abbreviations: PACE: Programs of All-Inclusive Care for the Elderly.

**Figure 3 pharmacy-08-00087-f003:**
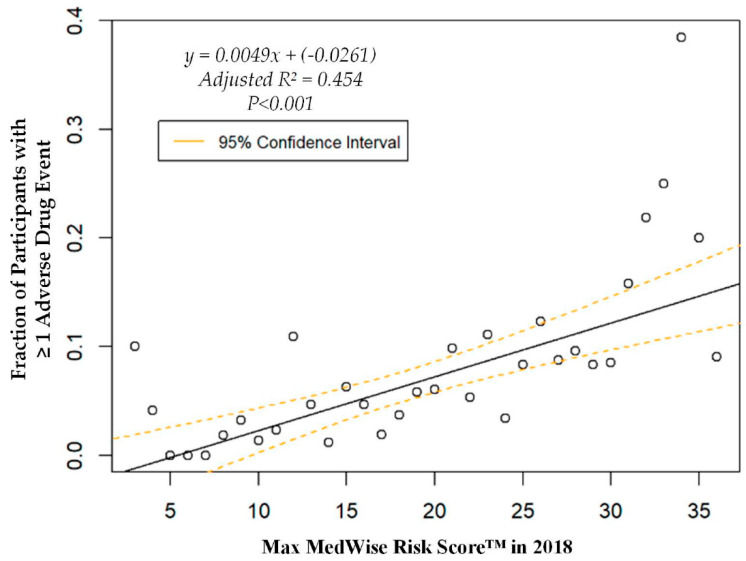
Weighted regression showing relationship between the MedWise Risk Score™ and participants with ≥1 adverse drug event.

**Figure 4 pharmacy-08-00087-f004:**
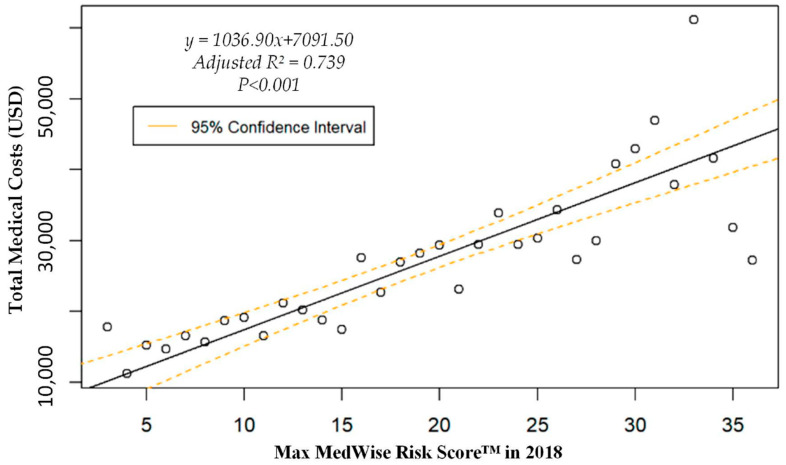
Weighted regression showing relationship between MedWise Risk Score™ and total annual (2018) medical costs in the total population.

**Figure 5 pharmacy-08-00087-f005:**
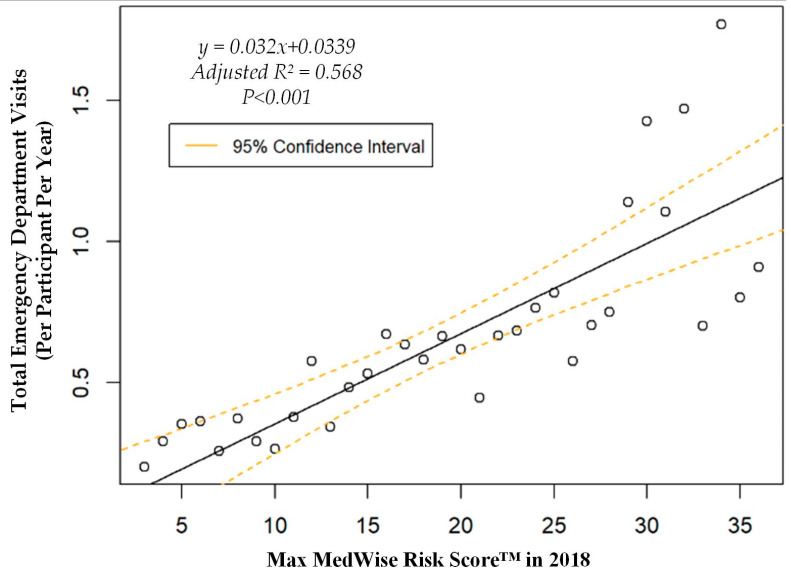
Weighted regression showing relationship between MedWise Risk Score™ and emergency department visits in the total population.

**Figure 6 pharmacy-08-00087-f006:**
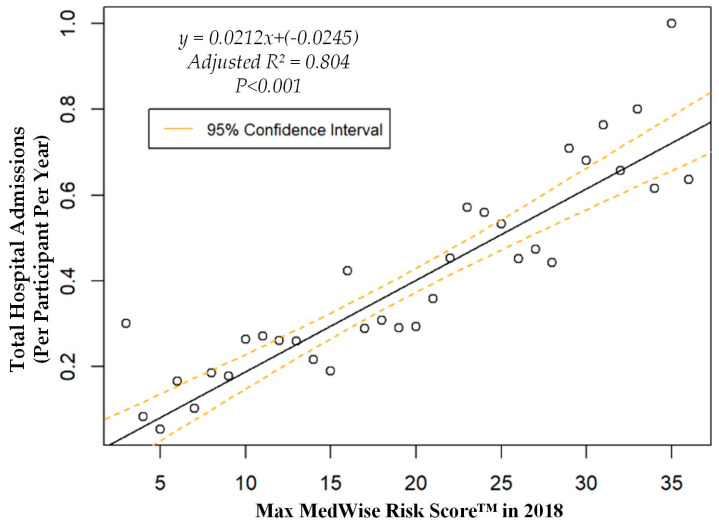
Weighted regression showing relationship between MedWise Risk Score™ and hospital admissions in the total population.

**Figure 7 pharmacy-08-00087-f007:**
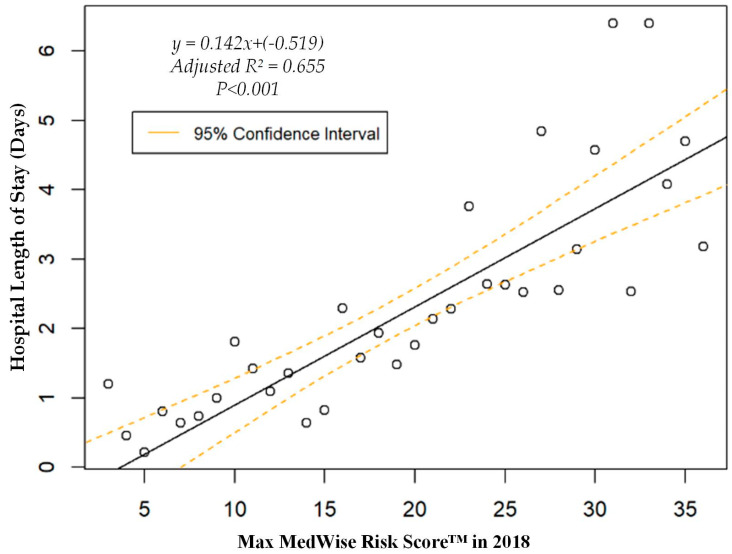
Weighted regression showing relationship between MedWise Risk Score™ and hospital length of stay in the total population.

**Table 1 pharmacy-08-00087-t001:** Overall study demographics.

Characteristic	Total	Participants without ADEs	Participants with ADEs
**PACE participants, *n* (%)**	1965 (100)	1837 (93.5)	128 (6.5)
Age, mean ± SD (range)	76.8 ± 9.9 (56–104)	77.1 ± 9.9 (56–104)	72.9 ± 9.7 (56–96)
Age, median (IQR)	77 (69, 84)	77 (69, 84)	71 (66, 80)
Female, *n* (%)	1282 (65.2)	1206 (65.7)	76 (59.4)
Number of profiled medications, mean ± SD (range) ^1^	13.4 ± 5.8 (1–42)	13.2 ± 5.8 (1–42)	16.2 ± 6.1 (4–37)
Number of profiled medications, median (IQR) ^1^	13 (9, 17)	13 (9, 17)	15.5 (12, 20)
MedWise Risk Score™, mean ± SD (range)	18.5 ±7.8 (2–40)	18.2 ± 7.7 (2–40)	23.1 ± 7.8 (3–40)
MedWise Risk Score™, median (IQR)	18 (12, 25)	18 (12, 24)	23 (17.75, 29)
**Ethnicity, *n* (%)**			
Non-Hispanic White	1140 (58.0)	1061 (57.8)	79 (61.7)
Black	454 (23.1)	422 (23.0)	32 (25.0)
Hispanic	142 (7.2)	135 (7.3)	7 (5.5)
Asian or Pacific Islander	59 (3.0)	58 (3.2)	1 (0.8)
American Indian/Alaskan Native	7 (0.4)	7 (0.4)	0 (0.0)
Unknown or other	163 (8.3)	154 (8.4)	9 (7.0)
**Top chronic comorbidities (by diagnosis code), *n* (%)**			
Essential (primary) hypertension	1233 (62.7)	1135 (61.8)	98 (76.6)
Type 2 diabetes mellitus without complications	790 (40.2)	727 (39.6)	63 (49.2)
Hyperlipidemia, unspecified	622 (31.7)	573 (31.2)	49 (38.3)
Atherosclerotic heart disease of native coronary artery without angina pectoris	428 (21.8)	388 (21.1)	40 (31.2)
Hypothyroidism, unspecified	421 (21.4)	385 (21.0)	36 (28.1)
Unspecified dementia without behavioral disturbance	406 (20.7)	374 (20.4)	32 (25.0)
Chronic obstructive pulmonary disease, unspecified	373 (19.0)	335 (18.2)	38 (29.7)
Chronic kidney disease, stage 3	371 (18.9)	335 (18.2)	36 (28.1)
Heart failure, unspecified	349 (17.8)	314 (17.1)	35 (27.3)
**Pertinent risk outcomes**			
Total medical expenditures, mean (95% CI), USD	$26,299.97 (24,917.54, 27,682.39)	$25,087.07 (23,720.95, 26,453.19)	$43,706.95 (36,149.34, 51,264.56)
Hospital admissions, mean (95% CI), *n* per participant per year	0.37 (0.33, 0.41)	0.34 (0.3, 0.37)	0.84 (0.62, 1.05)
Hospital length of stay, mean (95% CI), days	2.1 (1.8, 2.4)	1.9 (1.7, 2.2)	4.8 (3.3, 6.2)
ED visits, mean (95% CI), *n* per participant per year	0.63 (0.57, 0.69)	0.59 (0.53, 0.65)	1.21 (0.88, 1.54)
**PACE organizations, *n***	12		
Geographic location of PACE, *n* (%)			
Northeast	3 (25.0)		
South	4 (33.3)		
Midwest	1 (8.3)		
West	4 (33.3)		
**PACE census size, *n* (%)**			
<120 participants	1 (8.3)		
120–220 participants	7 (58.3)		
>220 participants	4 (33.3)		

Abbreviations: PACE: Programs of All-Inclusive Care for the Elderly; ADE: adverse drug event; ED: emergency department; IQR: interquartile range; CI: confidence interval. ^1^ Medication counts obtained using December 2018 data. To facilitate reading, subsections are bolded.

**Table 2 pharmacy-08-00087-t002:** Most common ^1^ adverse drug events (ADEs) identified by ICD-10 codes.

ICD Code	Description	*n* (%)
F11.2	Mental and behavioral disorders due to use of opioids: dependence syndrome	20 (11.8)
F11.9	Mental and behavioral disorders due to use of opioids: unspecified mental and behavioral disorder	12 (7.1)
D68.3	Hemorrhagic disorder due to circulating anticoagulants	11 (6.5)
F19.1	Mental and behavioral disorders due to multiple drug use and use of other psychoactive substances: harmful use	9 (5.3)
T50.9	Poisoning: other and unspecified drugs, medicaments, and biological substances	9 (5.3)
G24.0	Drug-induced dystonia	7 (4.1)
L27.0	Generalized skin eruption due to drugs and medicaments	6 (3.5)
T40.2	Poisoning by other opioids (includes morphine and codeine)	5 (2.9)
G21.1	Other drug-induced secondary parkinsonism	5 (2.9)
T45.5	Anticoagulants (poisoning)	5 (2.9)
F19.2	Mental and behavioral disorders due to multiple drug use and use of other psychoactive substances: dependence syndrome	5 (2.9)

Abbreviations: ICD-10: International Classification of Diseases and Health Related Problems, 10th revision. ^1^ Accounts for 55.3% (*n* = 94) of the total ADEs identified by ICD-10 codes. In total, 54 unique ICD-10 codes identified ADEs. These codes appeared 170 times during 2018 claims.

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
