# Peer review of "Association of a Novel Medication Risk Score with Adverse Drug Events and Other Pertinent Outcomes Among Participants of the Programs of All-Inclusive Care for the Elderly"

_pharmacy, 2020, doi:10.3390/pharmacy8020087_

Round 1

Reviewer 1 Report

I found this to be a very well written / well presented paper which will be of high interest to a wide range of practitioners and researchers. I really enjoyed reading it!

I had only two suggestions for improvement:

Aim / objective: the objective was described in quite a vague way ("such as") and would benefit from some tightening up to make it really clear exactly what you were looking at.

While I appreciate that references were given regarding MRS, and a detailed description of this is beyond the remit of the paper, I would have found it helpful to know how it works in practice. By this I mean: how do practitioners access the scores and how is patient data added into the system to generate the scores? (I assume the organisation buys a licence and somehow the data is sent to the system and the score somehow sent back but I wasn't totally clear). After some reading / watching YouTube videos I had a better idea but a brief sentence added to the manuscript regarding this would benefit the reader. It would also help to understand the potential applicability to the wider health setting.

Reviewer 2 Report

Overall summary

Overall, this is a well-conducted and well-written manuscript for a significant topic with important implications. The objective of the study was to examine the association between a novel medication risk prediction tool, the MedWise Risk Score (MRS) and adverse events (ADE) and ADE related outcomes. The authors found significant association between MRS and ADEs and report a positive correlation between the MRS and higher costs, hospital admissions, ED visits and hospital length of stay. Two major critiques are insufficient information on the novel MRS tool and it is unclear if the additional outcomes assessed were ADE related in this study.

Other comments/suggestions/clarifications:

  1. Physician and facility charges/expenditure – authors mention that billed charges were examined, why were adjudicated claims not used? Adjudicated claims amount may be more accurate reflection of the costs.
  2. Based on the methods, it seems to be a cross-sectional study, please add this detail in the methods.
  3. The description of MRS is not in sufficient detail. Additional details of the novel MRS are warranted in this publication. Eg. how many and which single-drug adverse event pairs were considered in the calculation of the score? Was an adverse event identified for this factor? How was the combined risk score derived from 5 different factors? How was the MRS Score validated? Was there any multi-collinearity present/identified among the 5 factors (eg between factor 1 and factor 4)? Please include an interpretation of the MRS score values.
  4. The authors utilized the highest reported MRS during the study period, were other strategies considered? Eg. use the MRS prior to the identified ADE.
  5. How were multiple ADEs during the year handled?
  6. How were potential confounding factors handled while examining the association between MRS and outcomes?
  7. It is suggested to present the patient characteristics for the two groups of people - with and without ADEs.
  8. Figure 3 present the results for patients with one or more ADEs, are the figures 4-7 reported for the total population or in the patients with one or more ADEs?
